# Prevalence of Early Chronic Kidney Disease and Main Associated Factors in Spanish Population: Populational Study

**DOI:** 10.3390/jcm8091384

**Published:** 2019-09-04

**Authors:** Carmen Expósito, Guillem Pera, Lluís Rodríguez, Ingrid Arteaga, Alba Martínez, Alba Alumà, María Doladé, Pere Torán, Llorenç Caballeria

**Affiliations:** 1Centro de Atención Primaria Sabadell Centro, Instituto Catalán de la Salud, 08021 Sabadell, Barcelona, Spain; 2Fundació Institut Universitari per a la recerca a l’Atenció Primària de Salut Jordi Gol i Gurina (IDIAPJGol), Unitat de Suport a la Recerca Metropolitana Nord, 08303 Mataró, Spain (G.P.) (L.R.) (I.A.) (A.M.) (P.T.); 3Centro de Investigación Biomédica en Red de Enfermedades Hepáticas y Digestivas (CIBEREHD), 28029 Madrid, Spain; 4Centro de Atención Primaria Santa Eulàlia, Instituto Catalán de la Salud, 08187 Santa Eulàlia de Ronçana, Barcelona, Spain; 5Centro de Atención Primaria La Llagosta, Instituto Catalán de la Salud, 08120 La Llagosta, Barcelona, Spain; 6Laboratori Clínic Metropolitana Nord. Hospital Germans Trias i Pujol, 08916 Badalona, Spain (A.A.) (M.D.)

**Keywords:** chronic kidney disease, prevalence, associated factors, albuminuria, obesity, arterial hypertension, type 2 diabetes

## Abstract

The aim of this study was to determine the prevalence of early chronic kidney disease (EKD) (stages 1 and 2) and the factors associated. This was a populational study including individuals from 18–75 years randomly selected from 18 Primary Healthcare centers in the area of Barcelonès Nord and Maresme (Catalunya, Spain). Variables: anamnesis, physical examination, blood pressure, and analysis. EKD was defined with by a glomerular filtration rate (GFR) ≥60 mL/min/1.73 m^2^ and albumin/creatinine ratio (ACR) ≥17 mg/g in men and ≥25 mg/g in women confirmed with two determinations. 2871 individuals: 43% men, mean age 55 years (19–75), 32.2% obese, 50.5% abdominal obesity, 21.1% hypertensive, and 10.6% diabetic. Prevalence of EKD: With one determination 157 individuals (5.5%), 110 men (9%) and 47 women (2.8%); with two determinations 109 individuals (3.8%), 85 men (7%), and 24 women (1.5%). Factors independently associated with the multivariate logistic regression model: Man (OR 3.35), blood pressure ≥ 135/85 mmHg (OR 2.29), BMI ≥ 30 kg/m^2^ (OR 2.48), glycemia ≥ 100 mg/dL (OR 1.73), smoker (OR 1.67) and age (OR 1.04). The prevalence varies if the diagnosis is established based on one or two analytical determinations, overestimated if only one determination is made and depends on the value chosen to define urine albumin excretion.

## 1. Introduction

Recent papers in the literature have clearly shown that chronic kidney disease (CKD) is one of the main causes of morbidity and mortality in Western countries. Disease progression to the final stages of the disease requiring substitutive treatment with dialysis or kidney transplant leads to an important reduction in the quality of life of these patients and is an enormous economic burden to the healthcare system [1]. In addition, the associated cardiovascular complications are even more frequent than evolution to terminal disease, making the prevention, early detection and treatment of this disease by non-specialist physicians should be a priority [2]. CKD is defined as a reduction in the glomerular filtration rate (GFR) to less than 60 mL/min/1.73 m^2^ or the presence of markers of renal damage (mainly albumin values greater than or equal to 30 mg/day) during three months or more, independently of the cause, with adverse consequences for health [3,4]. According to the current KDIGO (Kidney Disease: Improving Global Outcomes) classification there are five stages of CKD based on the GFR and the presence of albuminuria; stages 1 and 2 correspond to a GFR ≥ 60 mL/min/1.73 m^2^, and the presence of albuminuria corresponds to early disease (EKD) or the onset of kidney disease [3,5]. 

Most studies measured the global prevalence of CKD, mainly focused in the advanced, 3, 4 and 5, stages of the disease. These studies showed great variability in the prevalence of CKD because of the different design of the studies, the methodology used, the diagnostic criteria, and the lack of standardization of the laboratory values. Taking these factors into account, the prevalence of EKD ranges from 1 to 30% [6,7,8,9,10,11,12,13,14,15,16]. Although many studies have determined the presence of albuminuria, only few studies have focused in the prevalence of EKD. With the different methodologies used in studies to evaluate the presence of albuminuria, urine strip [17], excretion methods [18] or concentrations [5,7,19,20], 24-h [18] or random urine [5,7,17,19,20], as well as a unique albuminuria measurement, the prevalence of EKD varies from 2.29% [19] to 28.6% [7]. There is currently consensus that the best method to evaluate albuminuria and to reduce this variability is to calculate the albumin/creatinine ratio (ACR) in the first urine of the morning [21].

An ACR greater than or equal to 30 mg/g of creatinine is usually considered to determine the presence of pathological albumin levels in urine [3,4,22,23]. However, some guidelines and groups of experts lowered this value in this limit as 17 mg/g for men and 25 mg/g for women [24,25,26], since these values have the best correlation with urine albumin excretion less than 30 mg/day, which is the normal value for healthy individuals. In addition, these values minimize, in part, the underdiagnosis of EKD in men [27].

In order to develop programmes at a populational level and avoid the progression of EKD to a more advanced stages, and the risk of associated complications, it is necessary to determine the prevalence and the characteristics of the population affected as accurately as possible from the earliest stages of the disease (stages 1 and 2) because these stages already are an independent risk marker for the development of subclinical arteriosclerosis [5] and cardiovascular disease [28]. The aim of this study was to determine the prevalence of EKD (stages 1 and 2) and the factors associated with this disease.

## 2. Materials and Methods

### 2.1. Study Design and Population

The study was carried out using a sample selected for the “detection of liver diseases in the general population” [29]. A descriptive, transversal, multicenter, population-based study was designed including individuals ascribed to 18 Primary Healthcare Centers of the area of Barcelonès Nord and Maresme (Catalonia, Spain), which covers to population of 470,000 inhabitants.

The sample was randomly selected from the database of the Primary Care Information System (SIAP) which includes all individuals with national healthcare cards and is equivalent to the population census of Catalonia, Spain. This database includes all the individuals ascribed to a Primary Healthcare Centre of the zone, regardless of whether they have been attended or not. All the candidates were invited to participate by a telephone call. For subjects accepting to participate a visit was programmed with a trained nurse who performed the anamnesis, physical examination and basal blood analyses.

The inclusion criteria were: population of both genders from 18–75 years of age ascribed to the participating Primary Healthcare Centers, who voluntarily and provided written informed consent accepted to participate in the study.

The exclusion criteria were: CKD stages 3, 4 and 5; subjects with severe disease in advanced stage and/or clinically unstable (heart failure, chronic obstructive pulmonary disease or cancer) or conditions making data collection and follow-up difficult, such as incapacitating conditions, cognitive impairment or individuals in long-term care facilities.

The protocol was approved by the Ethics Committee of the Fundació Gol i Gorina (P11/58) (Barcelona, Spain) which followed the Declaration of Helsinki. All subjects provided written informed consent before inclusion.

### 2.2. Study Variables

#### 2.2.1. Sociodemographic Variables: Age, Gender

Age: Age in years in subjects between 18 and 75 yearsGender: Males and females

#### 2.2.2. Anamnesis

Presence of comorbidities. This was determined by review of the clinical history: arterial hypertension (AHT), type 2 diabetes mellitus (DM2), hypercholesterolemia and hypertriglyceridemia.Alcohol intake. The consumption of alcohol was recorded as standard drink units (SDU), and intake during the week and the weekends was differentiated. The length of consumption in years was also recorded. One SDU is equivalent to 10 g of alcohol. Consumption per week ≥ 21 SDU in men and ≥ 14 SDU in women was considered to be of risk.Tobacco consumption: This included never smokers, ex-smokers (more than 1 years without smoking) and active smokers.

### 2.3. Physical Examination

Anthropometric data: weight (kg), height (cm), abdominal obesity (cm) and body mass index (BMI) calculated according to the weight formula (kg weight/m^2^ height). Abdominal obesity was measured by determining the waist circumference using a metric measuring tape (waist circumference is considered the perimeter of the intermediate abdominal zone between the last costal arch and the iliac crest measured on a horizontal plane).

Determination of blood pressure (BP): this was performed with the subject in a seated position with a validated (OMRON M6 Comfort) blood pressure monitor. Three measurements were made separated by two minutes each, and the mean of the last two measurements was taken as the blood pressure value, discarding the first measurement.

### 2.4. Analytical Determinations.

Blood analyses were performed after 12 h of fasting and included the determination of: complete blood count, basal glycemia, glycosylated hemoglobin, total cholesterol, HDL-cholesterol, LDL-cholesterol, triglycerides, serum creatinine, GFR, and ACR in the first morning urine.

### 2.5. Diagnosis of EKD

According to the KDIGO guidelines, EKD is considered to be before stages 1 and 2 [3] and includes individuals with a GFR ≥ 60 mL/min/1.73 m^2^ and the presence of albuminuria. The GFR was first calculated using the MDRD-4 (Modification of Diet in Renal Disease-4) formula [30,31] which was used by the reference laboratory at the beginning of the study and afterwards, the new CKD-EPI (Chronic Kidney Disease Epidemiology Collaboration) formula [32] was implemented by the reference laboratory. The presence of albuminuria was determined using the ACR in the first urine of the morning. The cut-offs for defining the presence of pathological albuminuria were ≥17 mg/g in men and ≥25 mg/g in women. An altered value required confirmation with a second determination in no less than 3 months. In the case of discordance between the first and second urine albumin values (first sample altered and second normal), a third determination was made in no less than 3 months following the second measurement to definitively confirm or rule out the presence of albuminuria (Figure 1).

### 2.6. Statistical Analysis

For data collection, a notebook was designed to register each of the variables of anamnesis, physical examination, and the analytical results. A database was developed in an Excel spreadsheet, with all the variables being introduced with the double entry method. Exhaustive quality control of the data was performed.

The variables are expressed are frequencies and percentages when categorical and as mean and standard deviation (SD) when quantitative. However, clearly asymmetric values are shown as median and interquartile range (IQR).

Bivariate comparisons were carried out with the Chi-square test to compare categorical variables using the Fisher exact test when the expected frequency in some domain was less than 5. Continuous variables were compared with the Student’s t test, using the Mann-Whitney test if the data were expressed as medians.

The prevalences were calculated with their respective 95% confidence intervals (95%CI).

The multivariate relation between a dependent dichotomic variable (i.e., kidney disease: yes/no) and the potential explanatory and confounding factors were studied by logistic regression models obtaining odds ratios (OR) and their 95%CI adjusted for all the variables introduced in the model (shown in the tables). Variables showing a significant relationship in the bivariate models were included. When some explanatory variables were strongly correlated, one was chosen (that with the greatest biological plausibility or which had a greater effect on the bivariate model) to avoid collinearity.

All the statistical tests were performed bilaterally with a significance of 5%. The analyses were performed with the Stata v15 statistical package.

## 3. Results

The study was carried out from 27 March 2012 to 30 June 2016. The population from 18 to 75 years of age assigned to the participating primary care centers included 162.950 inhabitants. Of these, 3.460 individuals were randomly selected to initially be invited to participate in the detection of liver diseases in a healthy population, with 3.060 (62.8%) accepting to participate. After purification of the data for analysis, 22 individuals diagnosed with stages 3, 4 or 5 CKD and 167 in whom no urine sample was available were excluded. A total of 2.871 individuals were finally analyzed (Figure 2).

### 3.1. Characteristics of the Sample

Of the sample analyzed 1.222 individuals were men (43%) and 1.649 women (57%) (*p* < 0.001), with a mean age of 55 years (SD ± 12 years) (range: 19–75 years). Participation by age groups was as follows: 569 individuals (19.8%) between 19–44 years of age, 689 individuals (24%) between 45–54 years, 966 individuals (33.6%) between 55–64 years, and 647 individuals (22.5%) were in the 65–75 year age group (*p* < 0.001). Table 1 shows the main clinical characteristics of the study sample. Among the alcohol consumers (50%), 9.2% were drinkers of risk (*p* < 0.001). With respect to weight, 32.2% were obese, 41.6% presented overweight and 50.5% had abdominal obesity. Among the sample, 27.1% had hypertension, and 10.6% were diabetics.

### 3.2. Prevalence of EKD

In 157 individuals (5.5%) (110 men (9%) and 47 women (2.8%)) the GFR was ≥60 mL/min/1.73 m^2^, and the ACR was ≥17mg/g in men and ≥25 mg/g in women. Of these, 109 (3.8%) (85 men (7%) and 24 women (1.5%)) presented an ACR above the reference value in the second or third determinations (*p* < 0.001). Therefore, the global prevalence of EKD was 3.8% (Figure 3 and Figure 4). With an ACR cut-off of 30 mg/g for both genders, 103 individuals (3.6%) (73 men (6%) and 30 women (1.8%)) presented an ACR above this value in the first determination (*p* < 0.001), which was confirmed in 62 individuals (2.1%) (47 men (3.8%) and 15 women (1%), in the second or third determination (Figure 3 and Figure 4).

According to the different age groups, the prevalence of EKD was: 1.2% between 19 and 44 years, 2.2% between 45 and 54 years, 3.9% between 55 and 64 years and 7.6% between 65 and 75 years of age. The trend to an increase with age was also observed according to gender (*p* < 0.001) (Figure 5).

### 3.3. Principal Associated Factors

Table 1 shows the main comorbidities and clinical and analytical parameters associated with EKD. These were: AHT (8.4%; *p* < 0.001) and DM2 (15.4%; *p* < 0.001) registered in the clinical history of the patients, blood pressure (BP) ≥135/85 mmHg (6.4%; *p* < 0.001), glycemia ≥ 100 mg/dL (7%; *p* < 0.001), BMI ≥ 30 kg/m^2^ (6.7%; *p* < 0.001), abdominal obesity (4.9%; *p* = 0.001), triglycerides ≥150 mg/dL (6.8%; *p* < 0.001) and HDL-cholesterol <40 mg/dL in men and <50 mg/dL in women (5.1%; *p* = 0.086).

Figure 6 and Figure 7 show the prevalence of these factors and comorbidities in the 109 individuals with EKD.

The factors independently associated with EKD in the multivariate logistic regression model were: male gender (OR 3.35; 95%CI 1.98–5.68), BP ≥ 135/85 mmHg (OR 2.29; 95%CI 1.41–3.70), BMI ≥ 30 kg/m^2^ (OR 2.48; 95%CI 1.18–5.20), glycemia ≥100 mg/dl (OR 1.73; 95%CI 1.12–2.67), smoking habit (OR 1.67; 95%CI 1.02–2.73) and age (OR 1.04; 95%CI 1.02–1.07) (Table 2).

## 4. Discussion

The initial phases of CKD are silent and the prevalence is non negligible, having an important economic burden and social impact, especially in the end-stages. This has led to re-examination of the concept of this disease [33] and the development of populational programs for early detection and control by public health organisms to avoid comorbidities and progression to advanced phases. The prevalence of EKD in the present study was 3.8%. To our knowledge this is the first study on the detection of CKD in the early stages and the factors associated with the disease and in which the term “chronicity” is determined, confirming the persistence of the alteration of the kidney lesion after 3 months as recommended in the current guidelines of clinical practice [3,4].

Among the strengths of the present study, the following are of note: (1) the important number of subjects recruited, with a participation of 62.8% providing a representative sample of the general population of both genders, and (2) the populational character of the study with a totally random selection of participants made from the SIAP primary care database, which is even more up to date than the registry of the population census.

It is difficult to compare the prevalence observed in our study with other studies in the literature because of the important variability among studies in relation to not only the study populations but also the methodology used for the diagnosis of kidney disease [34,35]. Most of the epidemiological studies performed have evaluated the prevalence of CKD based on a single analytical determination of the GFR and albuminuria [6,7,8,9,10,11,12,14,15,16,19,36], despite current consensus that the diagnosis should be established on the presence of two altered analytical samples separated by at least 3 months [3,4], as was done in the present study, which mean an important difference with other published studies in general population and may result in significant variations in the prevalence of this entity. This is especially important in determined situations such as exercise, fever or systemic infections, cystitis or generalized inflammatory processes, among others, in which the permeability of the glomerular membrane may be transitorily increased to later return to its basal state. Confirming the presence of a chronic lesion with a second determination after a certain period of time avoids overdiagnosis due to these variations [36,37,38,39]. Although this study did not register these clinical situations, this may explain why an alteration of the ACR was observed in 5.5% of the sample and reduced to 3.8% when the measurement was repeated a second or third time 3 to 5 months later.

On the other hand, the prevalence of CKD also differs according to the cut-off established to define the presence of pathological albuminuria, and this was also observed in the present study. In most guidelines, including the KDIGO and the consensus statement of the Spanish Nephrology Society [3,4,22,23], this cut-off is an ACR ≥ 30 mg/g in both genders, considering that above this value the risk of global mortality, progression to CKD, cardiovascular disease and exacerbation of CKD is greater [40]. However, the relationship between albuminuria and the risk of renal and cardiovascular complications is a continuous variable [41,42]. If we accept a cut-off of 30 mg/g, the prevalence of EKD which we observed with a single ACR determination was 3.6%, decreasing to 2.1% with the second or third confirmatory determination. This prevalence is similar to that observed in the Spanish EPIRCE (Epidemiology of Chronic Renal Insufficiency in Spain) populational study in which 2.29% of the population presented EKD, although this result was obtained with a single ACR determination [19]. The prevalences reported in the NHANES questionnaire (The *National Health and Nutrition Examination Survey)* 1988–1994 and 1999–2004 were higher, with 4.4% and 5%, respectively [7], which may have been explained by the greater prevalence of factors associated with CKD in the American population. The prevalence described in the KHANES survey in the Korean population was also higher at 5.7%, but in this study, albuminuria was determined using the urine strip [43]. This wide variability in the prevalence of CKD has also been observed in countries of the European Union (EU), ranging from the 3.31% reported in Norway to 17.3% in north-east Germany. These differences remained after adjustment for diabetes mellitus, AHT and obesity, which are considered to be important risk factors of CKD [34]. The different study populations, genetic variability of the different ethnicities or in different territories and the heterogeneity of the laboratory methods for measuring and storing both serum creatinine and albuminuria [44,45] contribute to these differences.

The results of the prevalence of CKD according to gender are also controversial depending on the threshold value to define the presence of albuminuria. In the present study ACR values ≥ 17 in men and ≥25 mg/g in women, showed EKD to be present in 7% of men and 1.5% of women, being clearly higher in men. In the NHANES 1999–2000 study, the prevalence was higher among women and the EPIRCE study described a greater prevalence in men, but the differences were not significant. In both studies, the ACR cut-off was 30 mg/g for both genders. In the present study, this cut-off also showed a higher prevalence in men, being statistically significant regardless of whether a single or two ACR determinations were made. However, the use of this cut-off for both genders may underestimate the prevalence of kidney damage in men [19,46]. In normal conditions, the ACR value is higher in women since urine albumin excretion is practically the same in both genders, but creatinine excretion is greater in men because of the greater muscle mass they present [47]. Therefore, some scientific guidelines and societies [26,48,49] have established cut-off values ≥17 mg/g for men and ≥ 25 mg/g for women as indicative of the pathological presence of albuminuria, because these are the values which best correlate with the albumin excretion of 30 mg/d considered to be the maximum in healthy individuals [27].

Age is a clear factor of risk for the appearance of CKD in all its stages [7,34,35,43,46,48], and the present study showed an increase in the prevalence of EKD in the older age groups, being greater in men of any age group. This trend was also observed in other studies in Spain and other countries [5,7,19,34]. Despite this, some authors have questioned the clinical repercussion which the appearance of CKD may have with age in regard to both the progression of the disease and its associated complications. A reduction in the GFR is a physiological condition inherent to aging. Indeed, some studies have reported that the relative risks associated with the complications of CKD do not increase in more advanced age groups, but rather aging itself leads to underlying endothelial dysfunction which extends to the arterial territory throughout the organism, producing generalized arteriosclerosis which would explain this increase of risk with age. The presence of albuminuria is the translation of this dysfunction at a glomerular level [40,42].

Apart from gender and age, the most important risk factors determining the appearance and progression of CKD are DM2, AHT, and obesity, together with smoking [50]. Of note in the present study was the high prevalence of overweight and obesity observed, with up to 73% of the sample presenting a BMI ≥ 25 kg/m^2^. This result is consistent with the trend to an increase in obesity in Western countries in the last years [51,52] and explains, in part, the increase in EKD. The prevalence of EKD in our obese population was 6.7%, being significantly greater than the 1.2% found in individuals with normal weight and is similar to that of other countries in the EU [34]. At a pathophysiological level, the synthesis and release of fatty acids in obesity leads to a chronic state of mild inflammation which has a direct impact on renal damage and the appearance of the remaining metabolic alterations associated with insulin resistance and diabetes [53,54,55,56,57,58]. This is important and we must insist on the management of metabolic factors as well as appropriate therapeutic measures should be applied for their control, especially in obese patients who present a moderate or high risk of kidney disease [59].

Diabetes is another main factor associated with the development and progression of CKD. In the present study, the prevalence of EKD in the diabetic population was 15.4%, with a trend to increasing compared to previous studies [60], and placing these values within the mean of the EU in this population group [34]. Curiously, one study analyzing data from 1988 to 2012 in an American diabetic population described stability in the prevalance of EKD, decreasing from 23.1% in the period 1988–1994 to 17.2% in the period from 2011–2012, which is very similar to what was observed in the present study. The authors attributed these results to greater control of metabolic parameters in the last years [61].

Lastly, AHT is closely related to CKD and is one of the main factors of risk for the development of this disease. In our study not only AHT but also blood pressure values considered as normal-high doubled the risk of EKD after adjustment for different confounding factors, only being surpassed by obesity which increased the risk 2.5-old and was the most important risk factor for developing EKD in men. Among the population with hypertension in the present study, EKD was present in 8.4%, being within the range reported in the remaining countries of the EU [34].

Other metabolic alterations related to obesity such as low HDL or hypertriglyceridemia, which have shown an important relationship with the initial stages of kidney disease [5], were also found to be a risk factor in the present study, although this association was not statistically significant.

This study has several limitations: firstly, Although the selection of the sample was random, more women than men participated, due to a greater willingness to participate in the study. However, differences in gender distribution (43% men) are little compared with the real gender distribution among the population with the same age distribution in our area (47% men). Secondly, the calculation of the GFR was only performed based on one analytical determination which was not repeated at 3 months, implying stability of renal function. Some individuals may have been classified in an erroneous stage of kidney disease, but this would not have affected the results. On one hand, the analysis was performed in stages 1 and 2 together, and on the other hand, individuals erroneously classified in a stage lower than 1 or 2, and who were excluded from the study, represented a low percentage of the total sample. Thirdly, since the procedures were integrated in the usual clinical practice, the calculation of the GFR was initially performed using the MDRD-4 formula which was used by the reference laboratory and was later replaced by the CKD-EPI when this was implemented. This meant that stages 1 and 2 could not initially be differentiated, and some individuals were classified into different stages according to the formula used. Although MDRD-4 underestimates the GFR when this is above 60 mL/min/1.72 m^2^, a GFR lower than this value with both formulas is equivalent. Since the cut-off of the GFR used in the present study was 60 mL/min/1.73 m^2^, the results were not affected by the use of MDRD-4 or CKD-EPI. In addition, in most of the cases, the GFR calculated with MDRD-4 was recalculated with CKD-EPI. Fourthly, in determined situations such as the important loss of muscle mass, muscular diseases or morbid obesity, among others, the GFR cannot be calculated with the MDRD-4 or CKD-EPI, and therefore, 24-h urine clearance was the method of choice to evaluate kidney function in these cases. However, a very low percentage of these individuals were included in the study since most were excluded at the initiation of the study. Finally, although this study is population-based, note that it’s derived from a study intended to estimate the prevalence of silent hepatic fibrosis. As a consequence, people with known chronic liver disease were excluded from the original study. This accounts for a 1.2% of the sample. This could slightly infraestimate the prevalence of EKD since liver and chronic diseases are associated.

## 5. Conclusions

The prevalence EKD among the general population is not negligible. It is clearly higher among men and increases with age. This prevalence significantly differs if the diagnosis is based on one or two analytical determinations, being overestimated if only one measurement is made and depends on the cut-off value chosen to define albuminuria. EKD was most frequent among obese subjects, diabetics and individuals with hypertension, and the factors independently associated with the prevalence of EKD were male gender, obesity, and blood pressure. Nonetheless, the origin of CKD is multifactorial, and the variability among the methodologies used in different studies should be reduced in order to focus on the factors which influence, and to what extent, the development of kidney disease in different populations and to provide adequate therapeutic interventions.

## Figures and Tables

**Figure 1 jcm-08-01384-f001:**
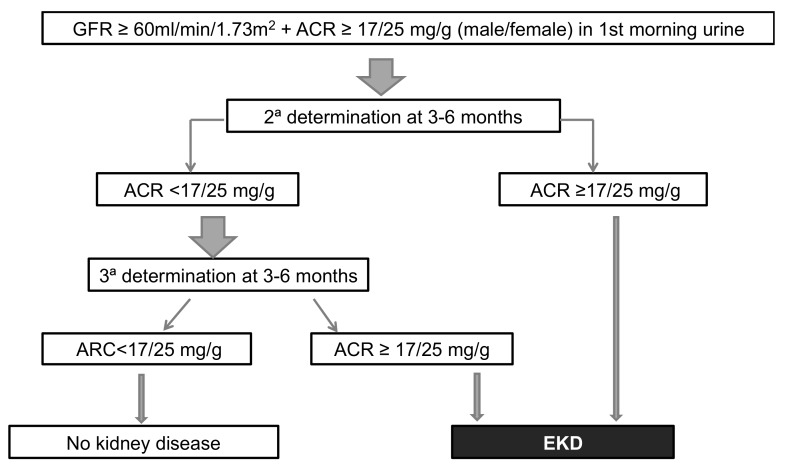
Diagnosis of early kidney disease (EKD).

**Figure 2 jcm-08-01384-f002:**
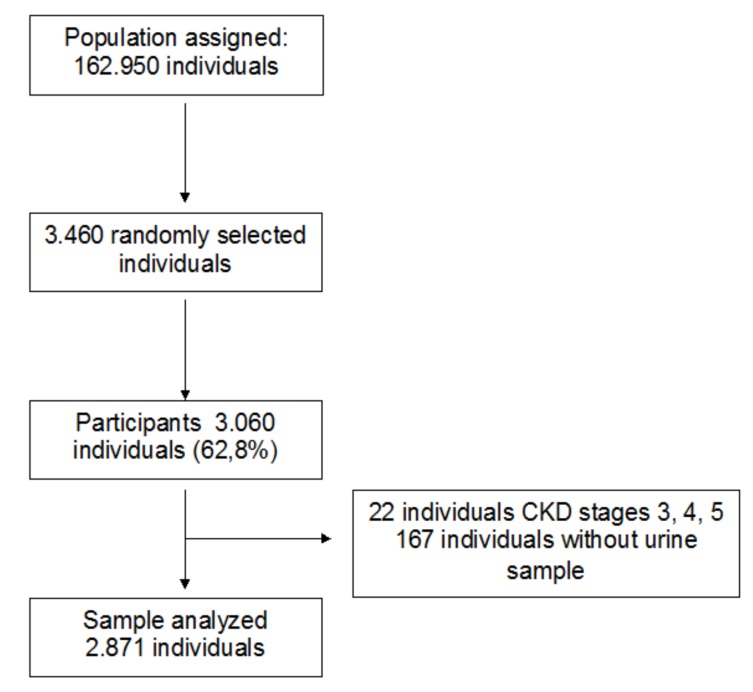
Diagram of participation.

**Figure 3 jcm-08-01384-f003:**
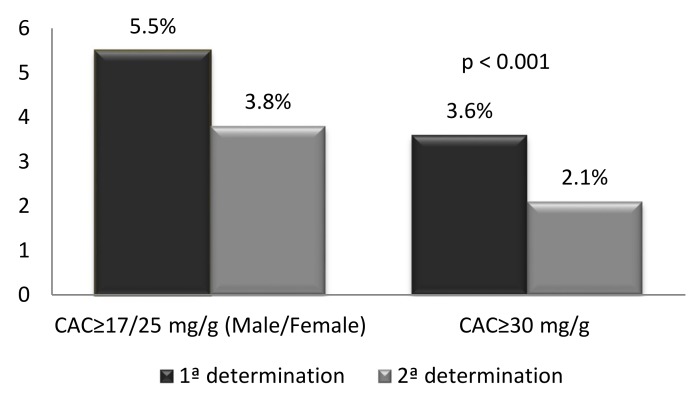
Prevalence of early kidney disease according to cut-off values.

**Figure 4 jcm-08-01384-f004:**
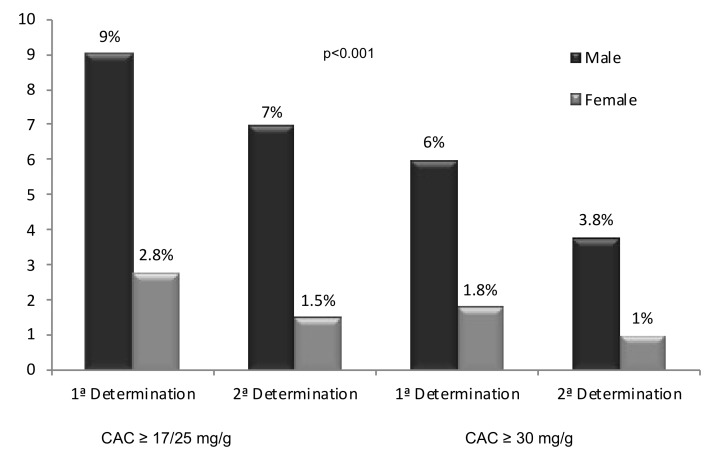
Prevalence of early kidney disease by gender.

**Figure 5 jcm-08-01384-f005:**
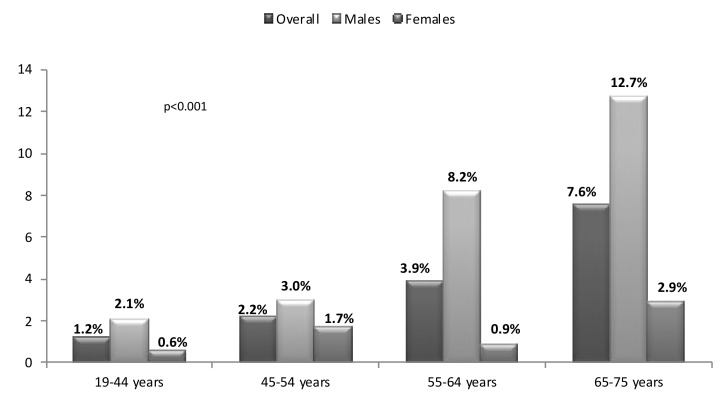
Prevalence of early kidney disease by age groups and gender.

**Figure 6 jcm-08-01384-f006:**
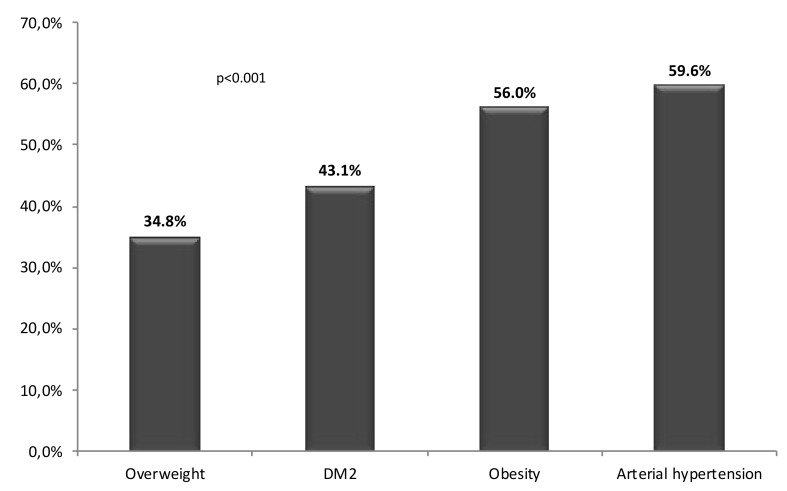
Prevalence of comorbidities in the population affected by early kidney disease.

**Figure 7 jcm-08-01384-f007:**
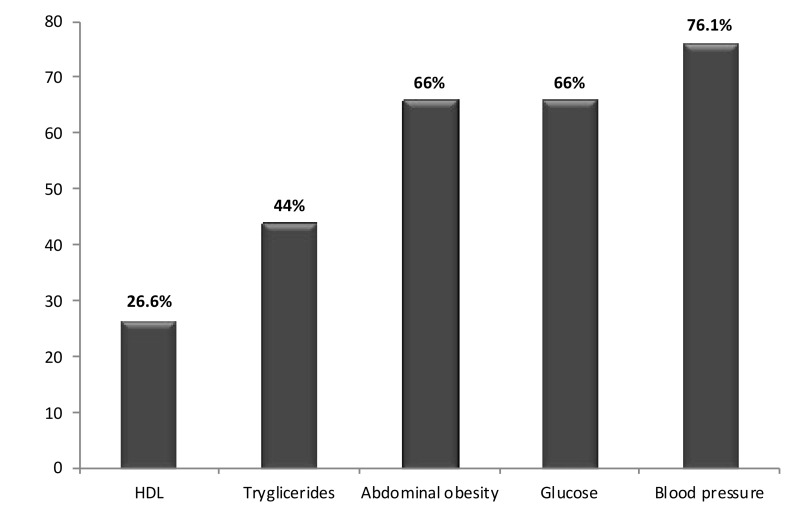
Prevalence of associated factors in the population with early kidney disease.

**Table 1 jcm-08-01384-t001:** Baseline characteristics of the 2871 subjects included in the study and associated factors according to the presence of early kidney disease (EKD).

Variables	EKD				Total		
	No		Yes				
	n	%	n	%	n	%	*p*
Gender							<0.001
Males	1137	93.0%	85	7.0%	1222	43.0%	
Females	1625	98.5%	24	1.5%	1649	57.0%	
Age							<0.001
19–44	562	98.8%	7	1.2%	569	19.8%	
45–54	674	97.8%	15	2.2%	689	24.0%	
5–64	928	96.%	38	3.9%	966	33.6%	
65–75	598	92.4%	49	7.6%	647	22.5%	
Mean, (±SD)	55	12	61	9	55	12	<0.001
Tobacco							<0.001
Non smoker	1339	98.0%	28	2.0%	1367	47.6%	
Former smoker	778	93.4%	55	6.6%	833	29.0%	
Current smoker	633	96.3%	24	3.7%	657	22.9%	
Alcohol							<0.001
Never drinker	1392	97.1%	42	2.9%	1434	50.0%	
Moderate drinker ^1^	1123	96.1%	45	3.9%	1168	40.6%	
Risk drinker	244	91.7%	22	8.3%	266	9.2%	
Obesity							<0.001
Normoweight (BMI < 25 kg/m^2^)	741	98.8%	9	1.2%	750	26.2%	
Overweight (25 ≤ BMI < 30 kg/m^2^)	1153	96.8%	38	3.2%	1191	41.6%	
Obese (BMI ≥ 30 kg/m^2^)	859	93.3%	62	6.7%	921	32.2%	
Abdominal obesity ^2^							0.001
No	1373	97.4%	36	2.6%	1409	49.5%	
Yes	1368	95.1%	71	4.9%	1439	50.5%	
Arterial hypertension							<0.001
No	2049	97.9%	44	2.1%	2093	72.9%	
Yes	713	91.6%	65	8.4%	778	27.1%	
Blood pressure (≥130/85 mmHg)							<0.001
No	1548	98.3%	26	1.7%	1574	55.0%	
Yes	1205	93.6%	83	6.4%	1288	45.0%	
Type-2 diabetes							<0.001
No	2504	97.6%	62	2.4%	2566	89.4%	
Yes	258	84.6%	47	15.4%	305	10.6%	
Glucose (≥100 mg/dL)							<0.001
No	1776	98.0%	37	2.0%	1813	63.7%	
Yes	963	93.0%	72	7.0%	1035	36.3%	
HDL < 40/50 mg/dL (Male/Female)							0.086
No	2184	96.5%	80	3.5%	2264	79.9%	
Yes	542	94.9%	29	5.1%	571	20.1%	
Triglycerides (≥150 mg/dL)							<0.001
No	2076	97.1%	61	2.9%	2137	75.3%	
Yes	653	93.2%	48	6.8%	701	24.7%	

^1^ Moderate drinker consumption per week <21 SDU in men and <14 in women. ^2^ Waist circumference ≥102 cm in men or ≥88 cm in women.

**Table 2 jcm-08-01384-t002:** Risk factors associated with early kidney disease.

	OR	IC95%		*p*
Age (per year)	1.04	1.02	1.70	0.001
Male gender	3.35	1.98	5.68	0.001
Smoker	1.67	1.02	2.73	0.042
Overweight (25 ≤ BMI < 30 kg/m^2^)	1.35	0.63	2.87	0.440
Obesity (BMI ≥ 30 Kg/m^2^)	2.48	1.18	5.20	0.016
Blood pressure (≥130/85 mmHg)	2.29	1.41	3.70	0.001
Fasting glucose (≥100 mg/dL)	1.73	1.12	2.67	0.013
Triglycerides (≥150 mg/dL)	1.51	1.00	2.29	0.053

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
