# Peer review of "Prevalence of Early Chronic Kidney Disease and Main Associated Factors in Spanish Population: Populational Study"

_jcm, 2019, doi:10.3390/jcm8091384_

Round 1

Reviewer 1 Report

Carmen et al. described the prevalence of early CKD from a population study in Spain. A total of 2871 excluding advanced CKD stage 3-5 and those with valid laboratories were investigated for 4 years. Of note, 43% men, 50.5% had abdominal obesity, and the results showed 5.5% had early CKD (110 men:9%, and 47 women: 2.8%). Of interest, the man had a much higher chance (OR 3.35) to have early CKD than women. The topic is interesting and meaningful, the study design is proper and the result is solid to answer the question. However, the manuscript still had some questions to be solved before considering to publication.

Q1:

Line 184: 3460 individual was randomly selected to be invited to participate…..

Please explicitly explain how to select randomly? To avoid selection bias.

Q2:

Line 191, men (43%) and women (57%), is a significant difference in gender after random selection, please explain!

Q3:

Table1: abdominal obesity comprised 50.5%, that means the non-abdominal obesity is 49.5%, but the level of significance is less than 0.05. why?

Q4:

Figure 3: cannot image why EKD prevalence is much lower in women, please explain

Q5: Table 1 is redundant, please use table 3 instead of table 1.

Q6: Please don’t use categorical classification in continuous variable such as blood pressure, glucose, HDL, TG, to avoid misclassification bias

Q7: Figure 6b, not easy to catch up on the idea, please delete

Q8: Table 4: use fasting glucose instead of glucose

Q9: Figure 7 showed be revised to show univariate logistic regression and multivariate logistic regression TOGETHER and how you select these factors in your model.

Q10: it is interesting in showing that obesity (BMI >30) had a higher probability to have early CKD than hypertension or type 2 diabetes. Can you give some ideas that abdominal obesity comprised nearly half in this study population? And the discussion in this part should be revised including citing more references including bariatric surgery in early CKD population (Effect of weight loss on the estimated glomerular filtration rates of obese patients at risk of chronic kidney disease: the RIGOR-TMU study. Lin YC, Lai YJ, Lin YC, Peng CC, Chen KC, Chuang MT, Wu MS, Chang TH.

J Cachexia Sarcopenia Muscle. 2019 Apr 2. doi: 10.1002/jcsm.12423. [Epub ahead of print])

Minor:

References were old, please cite the recent 10 years as possible. English should be reviewed again professionally from a native English speaker.

Author Response

Answer to Reviewer’s Comments on Manuscript jcm-570672

Reviewer 1

Carmen et al. described the prevalence of early CKD from a population study in Spain. A total of 2871 excluding advanced CKD stage 3-5 and those with valid laboratories were investigated for 4 years. Of note, 43% men, 50.5% had abdominal obesity, and the results showed 5.5% had early CKD (110 men:9%, and 47 women: 2.8%). Of interest, the man had a much higher chance (OR 3.35) to have early CKD than women. The topic is interesting and meaningful, the study design is proper and the result is solid to answer the question. However, the manuscript still had some questions to be solved before considering to publication.

Thanks for the comments issued

Q1:

Line 184: 3460 individual was randomly selected to be invited to participate…..

Please explicitly explain how to select randomly? To avoid selection bias.

In order to clarify this issue we have modified the text in the Material and Methods section including the type of randomization and the effort to contact as much of the individuals (lines 100-107, page 3).

The sample was randomly selected (simple randomization) from the database of the Primary Care Information System (SIAP) which includes all individuals with national healthcare cards and is equivalent to the population census of Catalonia, Spain. This database includes all the individuals adscribed to a Primary Healthcare Centre of the zone, regardless of whether they have been attended or not. All the candidates were invited to participate by a telephone call (up to 6 calls in different days/hours if the subject was not found). For subjects accepting to participate a visit was programmed with a trained nurse who performed the anamnesis, physical examination and basal blood analyses.

Q2:

Line 191, men (43%) and women (57%), is a significant difference in gender after random selection, please explain!

The real gender distribution in our population (with the same age distribution as in our study) is 47% men and 53% women. It’s true that in our sample women are overrepresented, due to a greater willingness to participate in the study. In order to clarify this issue we have added a comment in the discussion (limitations) to note this point (lines 357-360. Page 11).

Although the selection of the sample was random, more women than men participated, due to a greater willingness to participate in the study. However, differences in gender distribution (43% men) are little compared with the real gender distribution among the population with the same age distribution in our area (47% men).

Q3:

Table1: abdominal obesity comprised 50.5%, that means the non-abdominal obesity is 49.5%, but the level of significance is less than 0.05. why?

Although there is not a clinical significant difference, the statistical difference is high (p=0.001). This is due to the great sample available (>2800 subjects). Note that in the new version table 1 is deleted (and this information can be found in the former table 3, now Table 1).

Q4:

Figure 3: cannot image why EKD prevalence is much lower in women, please explain

Figure 3 shows the overall prevalence, both in men and women. We want to show what this prevalence is according to the cut-off values that we have taken in the present study: ≥17mg/g for men and ≥25mg/g for women, and compare them with the cut-off ACR ≥30mg/g. The reason for taking the cut-off values of 17 and 25 is explained in the introduction (lines 82-87) and in the discussion (lines 301 to 315).

What is incomplete and not understood is Figure 4. There are missing the cut-off points in the legend.

Q5: Table 1 is redundant, please use table 3 instead of table 1.

We agree with the reviewer and now former table 1 is deleted and former table 3 is now Table 1.

Table 2 has also been deleted

Q6: Please don’t use categorical classification in continuous variable such as blood pressure, glucose, HDL, TG, to avoid misclassification bias

We understand the concern of the reviewer using dichotomized risk factors. However, this is a very popular way of showing this variables, more reader-friendly than continuous values, using cut-points derived from de metabolic syndrome definition. In addition, although categorizing variables has known pitfalls, avoids other linked to continuous variables as outlier effects or potential departures from classical statistical assumptions. Note that the effect using continuous variables is comparable to those we show using categorized variables. So, unless the editor does not agree, we prefer to keep the current format of the variables.

Q7: Figure 6b, not easy to catch up on the idea, please delete

Sorry but we do not have any figure 6b in our paper. Please, let us know if this is a typo. Thank you.

Q8: Table 4: use fasting glucose instead of glucose

Changed.

Q9: Figure 7 showed be revised to show univariate logistic regression and multivariate logistic regression TOGETHER and how you select these factors in your model.

We guess the reviewer refers to table 4. We do not agree to include the results from univariate logistic regression, since they are not taking into account potential confounders. In addition, univariate logistic regression results can be deduced from current table 1 (former table 3) where characteristics of the participants with EKD are compared with those without EKD. How we decided the final multivariate model variables? This is already explained in the Statistical analysis section: Variables showing a significant relationship in the bivariate models were included. When some explanatory variables were strongly correlated, one was chosen (that with the greatest biological plausibility or which had a greater effect on the bivariate model) to avoid collinearity.

Q10: it is interesting in showing that obesity (BMI >30) had a higher probability to have early CKD than hypertension or type 2 diabetes. Can you give some ideas that abdominal obesity comprised nearly half in this study population?

It has already been commented in the discussion that the prevalence of obesity and overweight has increased in Western countries. This is probably why there is such a high proportion of subjects with abdominal obesity. In our area of influence, several population studies have been carried out to determine the prevalence of nonalcoholic fatty liver1 and peripheral arthropathy2 and the presence of abdominal obesity was very high, similar to the values found in the present study.

1Caballeria L, et al. Eur J Gastroenterol Hepatol. 2010 Jan;22(1):24-32.

2Alzamora et al. BMC Cardiovascular Disorders 2013, 13:11

And the discussion in this part should be revised including citing more references including bariatric surgery in early CKD population (Effect of weight loss on the estimated glomerular filtration rates of obese patients at risk of chronic kidney disease: the RIGOR-TMU study. Lin YC, Lai YJ, Lin YC, Peng CC, Chen KC, Chuang MT, Wu MS, Chang TH. J Cachexia Sarcopenia Muscle. 2019 Apr 2. doi: 10.1002/jcsm.12423. [Epub ahead of print])

Thank you for this comment. In order to clarify this issue we have added a comment in the discussion (lines 336-339. Page 11).

This is important and we must insist on the management of metabolic factors as well as appropriate therapeutic measures should be applied for their control, especially in obese patients who present a moderate or high risk of kidney disease [59].

Minor:

References were old, please cite the recent 10 years as possible. English should be reviewed again professionally from a native English speaker. 

Updated and reviewed.

Reviewer 2 Report

The study by Exposito et al is interesting as it aims to estimate the prevalence of early kidney disease (EKD) in a sample of population in Spain using two measurements of albumin to creatinine ratio (ACR) at least 90 days apart as well as uses lower cutoffs for the ACR of 17 mg/g for men and 25 mg/g for men instead of 30 mg/g as used in some previous studies. The authors note a higher prevalence of EKD using these cutoffs and suggest that using these cutoffs prevents under-diagnosis of EKD especially in men. The strengths of this study include using two measurements as well as comparing the prevalence using the standard cutoffs of 30 mg/g vs sex specific cutoffs of of 17 mg/g for men and 25 mg/g for men.

Overall, I feel it merits publications with minor edits as follows

Minor comments

There are several typo/spelling errors that need to be corrected. Examples include Page 2 line 73 “nethodology” instead of methodology Page 3, line 100 “population: to “population” Table 1 “smoker” instead of “smoker” Please correct the above and recheck all spellings Figure 1: Change Man/Woman to Male/Female Figure 1: What is the definition of “moderate drinking” in this study Please correct all percentages used with decimals in the study to include “.” Instead of “’,” before the decimal. Example Figure 6: change “34,8%” to “34.8%” etc. Please change the same in all such numbers. Any reason why the sample chose was the one used for “detection of liver diseases” and not a random sample of the population? Please elaborate

Author Response

Reviewer 2

The study by Exposito et al is interesting as it aims to estimate the prevalence of early kidney disease (EKD) in a sample of population in Spain using two measurements of albumin to creatinine ratio (ACR) at least 90 days apart as well as uses lower cutoffs for the ACR of 17 mg/g for men and 25 mg/g for men instead of 30 mg/g as used in some previous studies. The authors note a higher prevalence of EKD using these cutoffs and suggest that using these cutoffs prevents under-diagnosis of EKD especially in men. The strengths of this study include using two measurements as well as comparing the prevalence using the standard cutoffs of 30 mg/g vs sex specific cutoffs of of 17 mg/g for men and 25 mg/g for men.

Overall, I feel it merits publications with minor edits as follows

Minor comments

There are several typo/spelling errors that need to be corrected. Examples include Page 2 line o73 “nethodology” instead of methodology Page 3, line 100 “population: to “population” Table 1 “smoker” instead of “smoker” Please correct the above and recheck all spellings Figure 1: Change Man/Woman to Male/Female Figure 1: What is the definition of “moderate drinking” in this study Please correct all percentages used with decimals in the study to include “.” Instead of “’,” before the decimal. Example Figure 6: change “34,8%” to “34.8%” etc. Please change the same in all such numbers. Any reason why the sample chose was the one used for “detection of liver diseases” and not a random sample of the population? Please elaborate 

Thank you for the comments issued

. The various spelling errors were corrected.

. We have changed the definition of man and woman in all figures and tables

. Decimals have been changed to points in the percentages of the figures.

. The definition of moderate drinker at the bottom of the table has been added.

. This study is based on simple randomization of the adult general population. It’s true that the unique restriction was to have not known liver chronic disease. We excluded about 1,2% of the potential sample due to this restriction. We have included a note about this issue in the limitations in the reviewed version of the paper.

In order to clarify this issue we have added a comment in the discussion (limitations) (lines 379-382. Page 12).

Although this study is population based, note that it’s derived from a study intended to estimate the prevalence of silent hepatic fibrosis. As a consequence, people with known chronic liver disease were excluded from the original study. This accounts for a 1.2% of the sample. This could slightly infraestimate the prevalence of EKD since liver and chronic diseases are associated.
